

# Influencing factors and health risk assessment of microcystins in the Yongjiang river (China) by Monte Carlo simulation

Chan-Chan Xiao[1], Mao-Jian Chen[2], Fan-Biao Mei[1], Xiang Fang[1], Tian-Ren Huang[1], Ji-Lin Li[1], Wei Deng[1] and Yuan-Dong Li[1]

[1] Department of Experimental Research, Affliated Tumor Hospital of Guangxi Medical University, Nanning, China
[2] Department of Breast Surgery, Affiliated Tumor Hospital of Guangxi Medical University, Nanning, China

Corresponding authors
Wei Deng, dengwei@gxmu.edu.cn
Yuan-Dong Li, lyd641209@163.com

## ABSTRACT

The Yongjiang river is a large, shallow, hyper-trophic, freshwater river in Guangxi, China. To investigate the presence of microcystin-RR, microcystin-LR, and microcystin-YR (MC-RR, MC-LR, and MC-YR) in the Yongjiang river and describe their correlation with environmental factors, as well as, assess health risk using Monte Carlo simulation, 90 water samples were collected at three sample points from March to December 2017. Results showed that during the monitoring period, total concentrations of MC-RR (TMC-RR), MC-YR (TMC-YR), and MC-LR (TMC-LR) varied from 0.0224 to 0.3783 μg/L, 0.0329 to 0.1433 μg/L, and 0.0341 to 0.2663 μg/L, respectively. Total phosphorus (TP) content appeared to be related to TMC-LR and the total concentrations of microcystins (TMCs), while pH and total nitrogen (TN)/TP ratio appeared to be related to TMC-RR and TMC-YR, respectively. Using the professional health risk assessment software @Risk7.5, the risks of dietary intake of microcystins (MCs), including the carcinogenic risk and non-carcinogenic risk, were evaluated. It was found that the carcinogenic risk of MC-RR from drinking water was higher than MC-LR and MC-YR, and the presence of MCs would lead to high potential health risks, especially in children. The carcinogenic risk of MC-RR to children was $>1 \times 10^{-4}$, the maximum allowance level recommended by the US Environmental Protection Agency; as for adults, it was $>5 \times 10^{-5}$, the maximum allowance level recommended by the International Commission on Radiological Protection. The non-carcinogenic hazard index (HI) of MC-RR, MC-YR, and MC-LR increased successively, indicating that MC-LR was more hazardous to human health than MC-YR and MC-RR, but its HI was <1. This suggests that MCs pose less risk to health. However, it is necessary to strengthen the protection and monitoring of drinking water source for effective control of water pollution and safeguarding of human health.

## INTRODUCTION

Eutrophication of freshwater bodies can result in algal blooms, especially those caused by cyanobacteria. The algal toxins secreted from cyanobacteria are possibly harmful to plants, animals, and humans (*Holland & Kinnear, 2013*; *Cao et al., 2017*). So far, most of as-known 90 microcystins (MCs) have been isolated from species and strains of *Microcystis* (*Pham & Utsumi, 2018*). Among them, the most widely distributed are microcystin-LR (MC-LR), microcystin-RR (MC-RR), and microcystin-YR (MC-YR) (*Zegura, 2016*). These toxins are synthesized in the cells and released after cell rupture, finally appeared as MCs in the water source.

Cyanobacteria blooms exist in eutrophicated waters worldwide, so that MCs can be bioaccumulated by aquatic animals and reach human bodies. These would severely harm human health and cause illness or deaths. In 1975, the drinking water source in the small town of Pennsylvania was contaminated by *Microcystis*, which resulted in acute gastroenteritis for over half of the local population (*Keleti et al., 1979*; *Keleti & Sykora, 1982*; *Lippy & Erb, 1976*). In 1996, due to contamination by MCs occurring in a hemodialysis center in Brazil, 116 of 130 patients developed symptoms of blurred vision and nausea, and >50 individuals succumbed to mortality (*Pouria et al., 1998*). Studies on drinking water showed that with drinking ditch pond water containing MCs, the mortality rate of local people in Haimen and Fu Sui caused by hepatocellular carcinoma reached about 100/100,000, which was significantly higher than that of shallow wells or deep wells (20/100,000) (*Ueno et al., 1996*). In 2010, the International Agency for Research on Cancer (IARC) listed MCs as a "possible human carcinogen" (Group 2B) based on its potential carcinogenicity (*International Agency for Research on Cancer (IARC), 2010*).

Although it has been confirmed that MCs can cause acute and chronic damage to human bodies, the risk assessment of MC-LR, MC-RR, and MC-YR in the Yongjiang river of China remains lack of reports. The environmental conditions of water source are crucial in the concentration levels of toxins. However, the factors affecting the concentration levels of MCs (nutrient levels and climatic conditions) in Yongjiang have not yet been elucidated. Therefore, it is urgent to investigate the concentration and distribution of MCs in Yongjiang river as influenced by seasonal changes in water quality and the related parameters.

Owing to the limitations of conducting toxicological health risk assessments in a population, the Monte Carlo simulation (mathematical and logical model) has been widely used in recent years. It was used to understand the behavior of water systems by assuming different problems and systems, showing an advantage over experiments (*Clausen et al., 2017*; *Sasaki et al., 2017*). Moreover, the Monte Carlo simulation of uncertainties was applied in the risk assessment model by collecting limited samples to predict the overall situation. As a result, the risk uncertainty was expressed intuitively, in agreement with the order of the nature, which favors for a decision-making for risk managers and policymakers (*Paladino, Moranda & Seyedsalehi, 2017*; *Sasi, Yozukmaz & Yabanli, 2017*). The US Environmental Protection Agency (USEPA) has set the Monte Carlo simulation as a basic method in the risk analysis policy (*Moolenaar, 1996*). Currently, American Palisade has
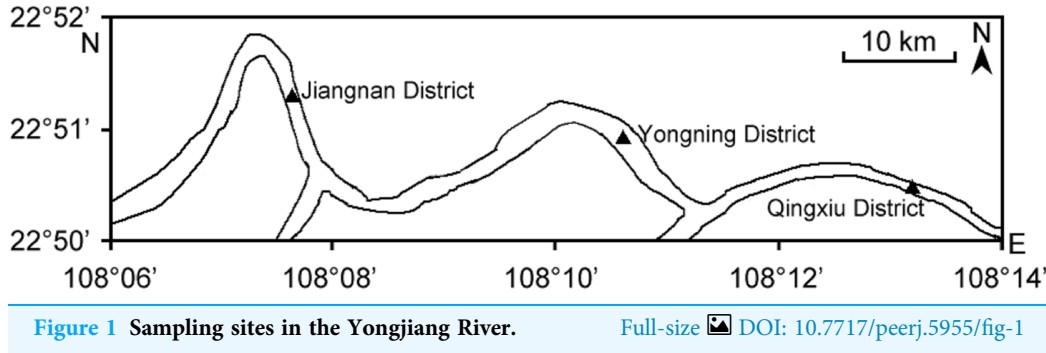

**Figure 1 Sampling sites in the Yongjiang River.**

developed the @Risk Monte Carlo software loading into Excel simulation technology for risk assessments. It is mainly based on the analysis of stochastic simulation method of Monte Carlo, which provides various predicted results by using a variety of probabilistic simulations, including the occurrence probability of events with a risk, forecasting the risk of uncertainty quantitatively, and summarizing the characterization results. For instance, *Li et al. (2017)* used the Monte Carlo model to assess the quantitative risk of aluminum in Youtiao, which did not exceed the provisional tolerable weekly intake set by the Joint Expert Committee on Food Additives for the public. *Jia et al. (2018)* used the Monte Carlo simulation to evaluate the trace elements in the four freshwater fishes from a mine-impacted river, and found that the consumption did not exert any appreciable adverse impact on human health due to the exposure to trace elements in fish muscle.

This study investigated the current status of drinking water sources in the Yongjiang river in China with respect to the contamination of MCs. The professional health risk assessment software @Risk7.5 was used to evaluate the risks of dietary intake of MCs, including the carcinogenic and non-carcinogenic risks. These findings provide a basis to develop an effective control of water pollution and quality in order to protect the human health in the specific area.

## MATERIALS AND METHODS

### Sampling location

The Yongjiang river is a major water resource with an average annual flow of 1,292 m$^3$/s in the Nanning City area (Fig. 1). The surface area of the river is 2,676 ha with a maximum depth of 23 m. It is the main urban water source in Nanning city, China, and the tributary channel is also a vital transportation route. Except for that, it is used for recreation as well as the source of water for domestic use, agriculture, fishery, and industry.

### Sampling

Water samples were collected from the Yongjiang river in Nanning City from March to December in 2017. For this, the river section was set into three sampling points: Qingxiu District, Jiangnan District, and Yongning District (Fig. 1). The water samples were collected at a depth of 0.5 m and three times/month from each sampling point. Thus, a total of 90 samples were collected during the most active daylight period (11.00–14.00) in 10 months. Sample collecting, containers, stabilization, and transportation to the

laboratory were in accordance with the methods described in *Wunderlin et al. (2001)*. Water samples were filtered through the 500 mesh stainless steel screens to remove large particles and were stored at 4 °C with the protection from light, finally processed within 24 h. A volume of 2,000 mL water sample was collected, and 500 mL water sample is passed through the 0.45 μm filter (Jinteng, Zhoushan, Zhejiang, China) under reduced pressure filtration. The filter containing the algae was subjected to an extraction process in order to recover the intracellular MCs (IMCs), followed by two extractions with five mL ultra-pure water after five times freezing-thawing at −80 °C/37 °C. After filtering through the 0.45 μm filter for removing the algal cells, the filtrates and the extracts from the filter were passed through solid phase extraction (SPE) (500 mg/6 mL) (SUPELCO, USA). The SPE was rinsed with 20 mL of 20% methanol and 10 mL deionized distilled water. The toxin was eluted from the stationary phase with 80% methanol (containing 0.05% Trifluoroacetic Acid (TFA)), and each sample was dried in a water bath under control temperature (60 °C).

## Water quality analysis

Water parameters $\chi^1$ = Water Temperature, $\chi^2$ = pH, $\chi^6$ = Dissolved Oxygen (DO) were measured in situ and $\chi^3$ = Total phosphorus (TP), $\chi^4 = PO_4^{3-} - P$, and $\chi^5$ = Total Nitrogen (TN) were measured in the laboratory. Each experiment was performed in triplicate, and the average values were reported. All water samples were analyzed using standard methods (*Guobiao (GB) Chinese National Standard, 2002*). The instruments used were YSI Model 58 thermometer, Knick Portamess 911 for pH measurement. DO using iodometric method. TN and TP were analyzed using Kjeldahl method and persulfate digestion. $PO_4^{3-} - P$ was determined according to stannous chloride method.

## Determination of MCs

The air-dried samples were suspended into one mL deionized distilled water for high performance liquid chromatography (HPLC) (LC-20A; Shimadzu, Kyoto, Kyoto Prefecture, Japan) analysis. The solubilized toxin samples were analyzed using HPLC with UV detector at 238 nm and symmetrical C18 column (3.9 × 150 mm) (Waters, Milford, MA, USA). The mobile phase consists of 33% acetonitrile and 67% deionized distilled water in 0.1% phosphate buffer (pH = 3.0). The flow rate was set at one mL/min. The injection volume was 20 μL and the column temperature was 45 °C. MC-LR and MC-RR (Solarbio, approximate purity, 95%) and MC-YR (Alexis, approximate purity 98%) were used as standards. Furthermore, the concentrations of MCs were determined by calibrating such area under the peak with the corresponding standard curves. MC-LR, MC-RR, and MC-YR showed a good linearity in the range of 0.025–2 μg/L ($r^2$ = 0.9987, 0.9992, 0.9997). Under the condition that the signal to noise ratio (S/N) is 3, the detection limits of MC-RR and MC-LR is 0.0125 μg/L, and the detection limit of MC-YR is 0.014 μg/L. The recoveries ranged from 91% to 110%, the relative standard deviation was 3.0–5.6%. A series of toxin peaks were identified using retention time and compared with spikes and known standards in the blank samples. Furthermore, the concentrations of MCs were determined by calibrating such area under the peak with corresponding standard curves. The order of the peaks and time of each standard substance were as

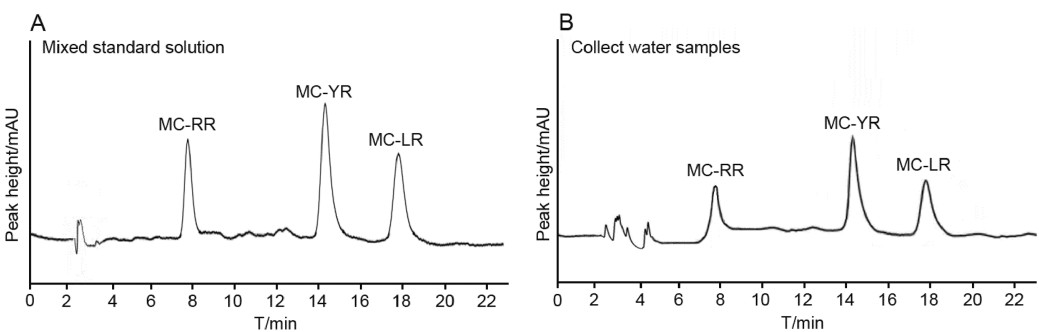

**Figure 2 HPLC chromatogram of MCs.** (A) HPLC chromatogram of MCs standards, the order of the peaks and time of each standard substance were as follows: MC-RR (7.599 min), MC-YR (14.225 min), and MC-LR (17.601 min). (B) HPLC chromatogram of MCs water samples, the same time represents the same substance, like A.

follows: MC-RR (7.599 min), MC-YR (14.225 min), and MC-LR (17.601 min). The test sample was analyzed in 18 min (Fig. 2).

## Method of risk assessment

### Construction of exposure assessment model

The three main routes of exposure to pollutants were consumption, inhalation, and skin absorption. The proportion of each varied in different pollutants. This study assessed the exposure risk caused by the drinking water. The daily exposure to MC-RR, MC-YR, and MC-LR in the drinking water was assessed using Monte Carlo simulation by @Risk7.5 software operating platform, while the Bootstrap sampling method was used for quantifying the uncertainty. Each Bootstrap sample was simulated with 10,000 Monte Carlo simulations to determine its uncertainty by obtaining different percentile values (P5–P95). The probabilistic assessment method was used to construct the exposure evaluation model (*Zobitz et al., 2011*).

The mechanism underlying the different exposures and various routes with different exposure dose formula were employed for the exposure assessment model of the health risk of chemical pollutants from the USEPA. Chronic daily intake (CDI) evaluated the safety of MCs in drinking water and the health risks of diverse routes in different populations. The CDI (μg/kg/day) formula for daily exposure of drinking water (*Duy et al., 2000*; *Funari & Testai, 2008*) is as follows:

$$CDI = \frac{Cw \times IR \times EF \times ED}{BW \times AT}$$

Here, Cw is the concentration of pollutants in the water, μg/L; IR is the ingestion rate of water (L/day); EF is the frequency of exposure, drinking water for daily necessities (365 days/year); ED is exposure duration (years); BW is the average body weight, kg; AT is the average time equal to ED multiplied by 365 days/year. According to the World Health Organization (WHO), the standard weight of adults is 70 kg, and the daily drinking volume is two L/day; while the weight of children is 16 kg and the daily drinking volume is 1 L/day (*World Health Organization (WHO), 2017*).

### Construction of risk description model

The characteristics of the pollutants in a water environment are generally divided into genetic, toxic substances (for instance, chemical carcinogens), and somatic toxic substances (for instance, non-carcinogens) as recommended by the USEPA water environmental health risk assessment model. The carcinogenic and non-carcinogenic risks of MCs in source waters are evaluated from exposure pathways.

(1) Health hazard risk model of chemical carcinogens

The formula of health hazard risk caused by chemical carcinogens recommended by USEPA (*Duy et al., 2000*):

$$R_i{}^c = \frac{1 - \exp(-D_i q_i)}{70}$$

In the formula, $R_i{}^c$ is the average personal carcinogenic annual risk of chemical carcinogen $i$ through drinking water, (years); 70 indicates the average life expectancy of Chinese population, years; $D_i$ is the daily average exposure to chemical carcinogen $i$ through drinking water, that is, CDI, μg/kg/day; $q_i$ is the carcinogenic strength coefficient of the chemical carcinogen $i$ through the drinking water, and currently, there is no recognized carcinogenic intensity coefficient of MCs. Based on the formula of carcinogenic strength coefficient of carcinogens (*Hitzfeld, Hoger & Dietrich, 2000*), this study deduced the formula as follows:

$$\text{CPI} = \frac{(\text{OR} - 1) \times \text{LR}}{D}$$

Where carcinogenic potency index (CPI) is the coefficient of carcinogenic strength estimated from the population data: $q_i$, kg/day/μg/L. Odds ratio (OR) refers to the ratio of the number of exposed and non-exposed people in the case group divided in the control group. According to the population of 80,000 inhabitants, the study showed that the person who drank the river water presented a liver cancer OR of 1.246 (*Falconer & Buckley, 1989*; *Yeh et al., 1989*; *Yu, Chen & Li, 1995*). Lifetime risk (LR) indicated the risk of cancer among individuals in the whole local population; according to the risk of cancer during the individual's lifespan exposed to MC-LR in the population of China, which was $6.2 \times 10^{-3}$ (*Yeh et al., 1989*; *Fan et al., 2009*). D indicated the calculation of the average daily life exposure dose, μg/kg/day. According to the study by Wolf et al., the lifetime carcinogenic strength of MC-RR is 1/10th of that of MC-LR, while the strength of carcinogenicity of MC-YR and MC-LR was equivalent (*Wolf & Frank, 2002*).

(2) Non-carcinogenic health risk assessment model

The health risk assessment model recommended by USEPA was used to evaluate the non-carcinogenic health risk of MCs in the Yongjiang river source water. The non-carcinogenic risk was described using a hazard index (HI) by the following formula:

$$\text{HI} = \frac{\text{CDI}}{\text{RfD}}$$

**Table 1 Results of analysis of MCs in water samples.**

| Toxin types | Number of samples | EMCs | IMCs | TMCs (EMCs + IMCs) concentration (µg/L) | |
| --- | --- | --- | --- | --- | --- |
| | | Detection Rate (%) | Detection Rate (%) | Range | Mean ± SD |
| MC-RR | 90 | 74.44 | 77.78 | 0.0224–0.3783 | 0.0727 ± 0.0599 |
| MC-YR | 90 | 64.44 | 56.67 | 0.0329–0.1433 | 0.0424 ± 0.0376 |
| MC-LR | 90 | 77.78 | 76.67 | 0.0341–0.2663 | 0.0763 ± 0.0637 |

Note:
The total concentrations of MCs (TMCs) in water is the sum of the concentrations of extracellular MCs (EMCs) and intracellular MCs (IMCs) dissolved in the water; standard deviation means SD.

Here, reference dose (RfD) is the RfD for MCs: the internationally accepted tolerable daily intake instead of MC-LR RfD was 0.04 µg/kg/d. According to the equivalent toxicity relationship among MC-RR, MC-YR, and MC-LR, the RfD values of MC-RR and MC-YR were 0.4 and 0.04 µg/kg/d, respectively (*Wolf & Frank, 2002*; *Lee et al., 2017*).

The HI is usually used as a benchmark of danger: HI > 1 indicates that the exposure higher than the RfD is harmful to the human body; HI ≤ 1 indicates that the exposure level is lower than the RfD, which is unlikely to be detrimental (*Younes, 1999*).

## Statistical analysis

The IBM SPSS Statistics 22.0 software was used to perform all descriptive statistical analysis, including minimum value, maximum value, mean value, standard deviation, Pearson's correlation analysis, and stepwise multiple linear regression. Moreover, the risk assessment of MC-RR, MC-YR, and MC-LR exposure in water source was carried out using @Risk7.5 probabilistic evaluation software based on the Monte Carlo simulation technique.

## RESULTS

### Concentration level, distribution characteristics, and environmental impact factors of MCs in source water

#### Concentration distribution characteristics of MCs in source water

The concentrations of MC-LR, MC-RR, and MC-YR in water samples were detected. The total concentrations of MCs (TMCs) in water are the sum of the concentrations of extracellular MCs (EMCs) and IMCs dissolved in the water. The results were summarized in Table 1.

#### Seasonal distribution characteristics of MCs in water samples

The seasonal distribution of MCs in the Yongjiang river is shown in Fig. 3.
The concentration of TMC-YR (referring to the sum of intracellular and extracellular, the same to the other twos) was significantly lower than the other two MCs. The concentrations of TMC-RR and TMC-YR at the same time reached the maximum levels in October.

#### Pearson's correlation analysis of environmental factors and MCs' concentration

Pearson's correlation analysis of environmental factors and MCs' concentration were analyzed and the results were shown in Table 2. According to the correlation analysis,

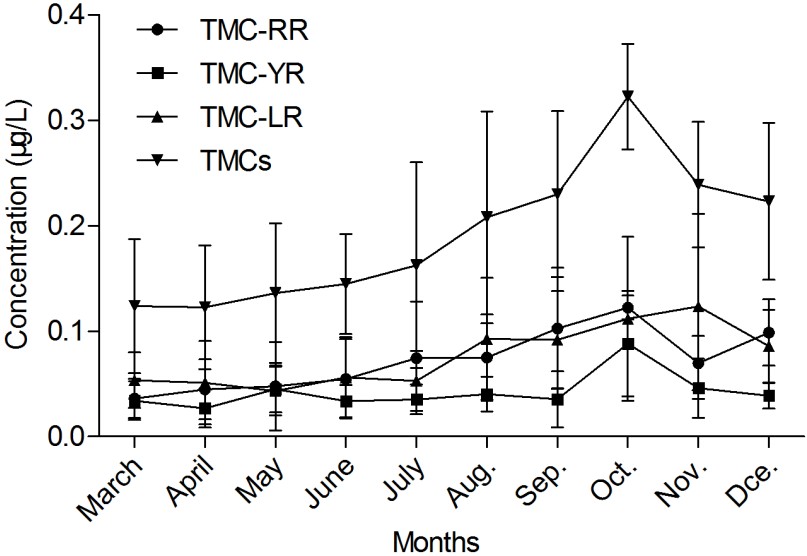

**Figure 3 Concentration of MCs in source water in various seasons.** TMCs is the sum of the total concentration (intracellular and extracellular) of each individual microcystin detected. This means that TMCs = TMC-LR + TMC-RR + TMC-YR.

**Table 2 Correlation coefficients between MCs and the influencing factors.**

| Environmental Factors | Correlation Coefficient | | | |
|---|---|---|---|---|
| | TMC-RR | TMC-YR | TMC-LR | TMCs |
| Water temperature | 0.436** | 0.085 | 0.480** | 0.614** |
| PH | −0.729** | −0.029 | −0.164 | −0.566** |
| DO | −0.063 | −0.076 | −0.768** | −0.570** |
| TN | 0.286** | −0.179 | −0.079 | 0.055 |
| TP | 0.043 | 0.073 | 0.851** | 0.610** |
| $PO_4^{3-} - P$ | −0.047 | −0.071 | 0.399** | 0.204 |
| TN:TP ratio | 0.040 | −0.229* | −0.434** | −0.347** |

Notes:
* Correlation is significant at the 0.05 level (two-tailed).
** Correlation is significant at the 0.01 level (two-tailed).

the concentration of TMC-RR was positively correlated with water temperature and TN ($p < 0.01$) with a significant negative correlation with pH ($p < 0.01$). The concentration of TMC-YR was negatively correlated with TN:TP ratio ($p < 0.05$). The concentration of TMC-LR was positively correlated with water temperature, TP, and $PO_4^{3-} - P$ ($p < 0.01$) with a significant negative correlation with DO and TN:TP ratio ($p < 0.01$). The concentration of TMCs was positively correlated with water temperature and TP ($p < 0.01$) with a significant negative correlation with pH, DO, and TN:TP ratio ($p < 0.01$).

### *Stepwise multiple linear regression analysis of MCs' concentration and environmental factors*

A stepwise multiple linear regression analysis of MCs' concentration and environmental factors is shown in Table 3. The results indicated that TP ($\chi^3$) is the dominant factor

**Table 3  Results of stepwise multiple linear regression.**

| MCs | Fitting equation | Correlation coefficient R | Adjusted $R^2$ | Tolerance | VIF | Model fitting | | Partial regression coefficient test | D–W statistic |
|---|---|---|---|---|---|---|---|---|---|
| | | | | | | **F** | **P** | | |
| TMCs | $y = 116.848 + 46.449\chi^3$ | 0.581 | 0.337 | 1 | 1 | 44.792 | <0.001 | $\chi^3(t = 6.693, P < 0.001)$ | 1.979 |
| | $y = 805.526 + 42.979\chi^3 - 92.886\chi^2$ | 0.716 | 0.513 | 0.989 | 1.001 | 45.87 | <0.001 | $\chi^3(t = 7.148, P < 0.001)$ $\chi^2(t = -5.608, P < 0.001)$ | |
| | $y = 538.069 + 35.557\chi^3 - 75.968\chi^2 + 6.452\chi^1$ | 0.741 | 0.549 | 0.693 | 1.443 | 34.938 | <0.001 | $\chi^3(t = 5.495, P < 0.001)$ $\chi^2(t = -4.396, P < 0.001)$ $\chi^1(t = 2.622, P = 0.01)$ | |
| TMC-RR | $y = 799.874 - 99.263\chi^2$ | 0.718 | 0.515 | 1 | 1 | 93.396 | <0.001 | $\chi^2(t = -9.664, P < 0.001)$ | 1.828 |
| | $y = 825.53 - 101.047\chi^2 - 35.61\chi^4$ | 0.738 | 0.544 | 0.994 | 1.006 | 51.98 | <0.001 | $\chi^2(t = -10.066, P < 0.001)$ $\chi^4(t = -2.375, P = 0.02)$ | |
| | $y = 676.095 - 91.169\chi^2 - 39.668\chi^4 + 3.42\chi^1$ | 0.759 | 0.576 | 0.846 | 1.181 | 38.908 | <0.001 | $\chi^2(t = -8.682, P < 0.001)$ $\chi^4(t = -2.709, P < 0.001)$ $\chi^1(t = 2.522, P = 0.014)$ | |
| TMC-YR | $y = 1.54 - 0.013\chi^7$ | 0.268 | 0.072 | 1 | 1 | 6.83 | 0.011 | $\chi^3(t = -2.613, P = 0.011)$ | 2.299 |
| TMC-LR | $y = 11.216 + 41.829\chi^3$ | 0.806 | 0.650 | 1 | 1 | 163.585 | <0.001 | $\chi^3(t = 12.79, P < 0.001)$ | 1.461 |
| | $y = 63.349 + 32.481\chi^3 - 6.653\chi^6$ | 0.848 | 0.718 | 0.678 | 1.476 | 111.004 | <0.001 | $\chi^3(t = 9.06, P < 0.001)$ $\chi^6(t = -4.592\ P < 0.001)$ | |

**Note:**

The fitting result of TMC-YR after lg10 conversion; D–W statistic (Durbin–Watson statistic); Variance Inflation Factor (VIF).

affecting the contents of TMC-LR and TMCs. pH ($\chi^2$) and TN:TP ratio ($\chi^7$) are the primary factors affecting the content of TMC-RR and TMC-YR, respectively. These findings were in agreement with the results of correlation analysis.

## Assessment of MCs' exposure in source waters

### Distribution fitting of the concentration of MCs in source water

The @Risk7.5 software is used to fit the processed samples. The concentration of MCs in source water were characterized as continuous data. The fitting results were followed by the optimal fitting distribution models: Gamma, Invgauss, Lognorm, Expon, and Loglogistic. Three main methods were used to test the goodness of the fittings: the Chi-Sq (Chi-squared) test, the K–S (Kolmogorov–Smirnov) test, and the A–D (Anderson–Darling) test (*Lipton et al., 1995*; *Cummins et al., 2009*). Above all, the sample fitting results (Table 4) were used to determine the fitting distribution types of the optimal probability of the pollution data: MC-RR had Gamma and Invgauss distribution, MC-YR had Lognorm, Expon, and Loglogistic distribution, and MC-LR had Gamma and Expon distribution. Furthermore, the results of the distribution parameters after fitting and the comparison with the sample data parameters are summarized in Table 5. The probability distribution of the mass concentration of MCs, MC-RR, MC-YR, and MC-LR in source water is shown in Figs. 4–6. The fitting results can be visually observed from the coincidence of the blue rectangular shape and the area under the red curve.
**Table 4 Fitting distribution and related parameters of MCs in source water (µg/L).**

| MCs | Fitting of Distribution | Distributed parameters | | Fit test sort | | | 50% Confidence value | 90% Confidence value | 95% Confidence value |
|---|---|---|---|---|---|---|---|---|---|
| | | Mean | Std Dev | K-S | A-D | Chi-Sq | | | |
| MC-RR | Gamma | 0.073 | 0.061 | 1 | 1 | 1 | 0.056 | 0.154 | 0.194 |
| | Invgauss | 0.073 | 0.066 | 5 | 2 | 2 | 0.053 | 0.154 | 0.202 |
| MC-YR | Lognorm | 0.047 | 0.071 | 4 | 2 | 1 | 0.026 | 0.103 | 0.154 |
| | Expon | 0.042 | 0.039 | 2 | 1 | 4 | 0.030 | 0.093 | 0.120 |
| | Loglogistic | 0.081 | – | 1 | 4 | 6 | 0.026 | 0.124 | 0.216 |
| MC-LR | Gamma | 0.076 | 0.070 | 1 | 1 | 2 | 0.055 | 0.168 | 0.216 |
| | Expon | 0.075 | 0.073 | 2 | 2 | 1 | 0.053 | 0.170 | 0.221 |

*Daily exposure calculation*

The @Risk7.5 software was utilized for the random extraction of the MCs concentration profiles from the water to calculate the daily exposure of direct drinking water by different populations to MC-RR, MC-YR, and MC-LR. Each simulation cycle was performed for 10,000 cycles, and the simulation results are shown in Table 6. A significant difference between adults and children was observed in the daily exposure. P50, P85, P90, and P95 (Table 6) represent the high exposure sites of each population. The MCs exposed to drinking water showed that the children's daily intake was twofold higher than that of the adults, suggesting that children are more susceptible to the pollution of MCs than adults.

## Risk characterization of MCs in source water

*Carcinogenic risk of MCs in source water*

Based on the exposure parameters and carcinogenic risk formula, @Risk7.5 risk analysis software was used to extract the numerical the value of MCs concentration in water randomly and calculate the carcinogenic risk of MC-RR, MC-YR, and MC-LR intake by different groups of individuals through direct drinking water. Each simulation cycle of 10,000 displayed the statistical simulation results summarized in Table 7. The carcinogenic annual risk of MC-YR was less than that of MC-LR and MC-RR, and MC-RR was the primary hazard in the source water. The maximum acceptable level (MAL) and the negligible level of the carcinogenic risk for the population recommended by some institutions are listed in Table 8; the annual risk in carcinogenesis of MCs in a water source is $10^{-6}$–$10^{-4}$ (*NHMRC & NRMMC, 2011*). The carcinogenic risk of MC-YR and MC-LR in adults and children was lower than the maximum acceptable risk level designated by USEPA ($1 \times 10^{-4}$) and the International Commission on Radiological Protection (ICRP) ($5 \times 10^{-5}$), and the risk of carcinogenesis in children was higher than that in adults. The health risks caused by MC-RR from drinking water source for children was significantly higher than the maximum allowance level recommended by USEPA ($1 \times 10^{-4}$). Similarly, the health risks caused by the MC-RR from drinking water source for adults were significantly higher than the maximum allowance level recommended by ICRP ($5 \times 10^{-5}$). These statistical details indicated that MC-RR in water bodies exhibited a significant carcinogenic risk to the health of adults and children.

**Table 5 Estimated value of the quantile for overall sample in different theoretical distributions.**

| Projects | Pollutant | Real value (μg/L) | Types | Predicted value (μg/L) | Relative difference of real value (%) |
|---|---|---|---|---|---|
| P50 | MC-RR | 0.058 | Gamma | 0.056 | 3.01 |
| | | | Invgauss | 0.053 | 7.81 |
| | | | Lognorm | 0.026 | 8.82 |
| | MC-YR | 0.024 | Expon | 0.030 | 25.17 |
| | | | Loglogistc | 0.026 | 10.03 |
| | MC-LR | 0.057 | Gamma | 0.055 | 2.02 |
| | | | Expon | 0.053 | 6.10 |
| P75 | MC-RR | 0.011 | Gamma | 0.099 | 13.00 |
| | | | Invgauss | 0.095 | 16.91 |
| | | | Lognorm | 0.053 | 13.88 |
| | MC-YR | 0.062 | Expon | 0.057 | 8.17 |
| | | | Loglogistc | 0.056 | 9.41 |
| | MC-LR | 0.112 | Gamma | 0.104 | 6.79 |
| | | | Expon | 0.104 | 7.46 |
| P90 | MC-RR | 0.136 | Gamma | 0.154 | 12.93 |
| | | | Invgauss | 0.154 | 13.09 |
| | | | Lognorm | 0.103 | 4.31 |
| | MC-YR | 0.108 | Expon | 0.093 | 14.10 |
| | | | Loglogistc | 0.124 | 15.12 |
| | MC-LR | 0.166 | Gamma | 0.168 | 1.09 |
| | | | Expon | 0.170 | 2.46 |
| P95 | MC-RR | 0.167 | Gamma | 0.194 | 16.26 |
| | | | Invgauss | 0.202 | 21.06 |
| | | | Lognorm | 0.154 | 29.08 |
| | MC-YR | 0.119 | Expon | 0.120 | 0.58 |
| | | | Loglogistc | 0.216 | 80.78 |
| | MC-LR | 0.215 | Gamma | 0.216 | 0.26 |
| | | | Expon | 0.221 | 2.56 |
| P99 | MC-RR | 0.378 | Gamma | 0.285 | 24.71 |
| | | | Invgauss | 0.321 | 15.02 |
| | | | Lognorm | 0.325 | 127.11 |
| | MC-YR | 0.143 | Expon | 0.183 | 27.56 |
| | | | Loglogistc | 0.735 | 412.77 |
| | MC-LR | 0.266 | Gamma | 0.326 | 22.62 |
| | | | Expon | 0.338 | 27.03 |

### Non-carcinogenic risk of MCs in source water

The exposure parameters and non-carcinogenic hazards index formula were used to calculate the values of different populations through direct drinking water intakes of MC-RR, MC-YR, and MC-LR (Table 9). These findings demonstrated that the average non-carcinogenic hazards index of MCs in different populations through drinking water

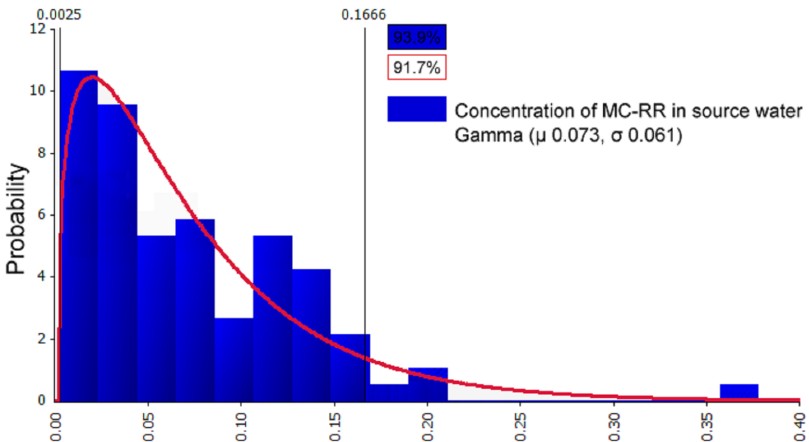

**Figure 4 Probability distribution graph after fitting of MC-RR in source water (µg/L).** The data comparison revealed that the optimal fitting distribution of the most suitable concentration of MC-RR in source water was Gamma (µ 0.073, σ 0.061) (first number µ as the position parameter and the second number σ as the scale parameter). The abscissa in Fig. 4 represents the concentrations of MC-RR; the concentrations are partitioned, the length of each interval is the group distance, the rectangular area is the frequency of the group, and the ratio of the total sample utilized, and the vertical axis is the frequency divided by the group distance obtained. The fitting results can be visually observed from the coincidence of the blue rectangular shape (93.9%) and the area under the red curve (91.7%).

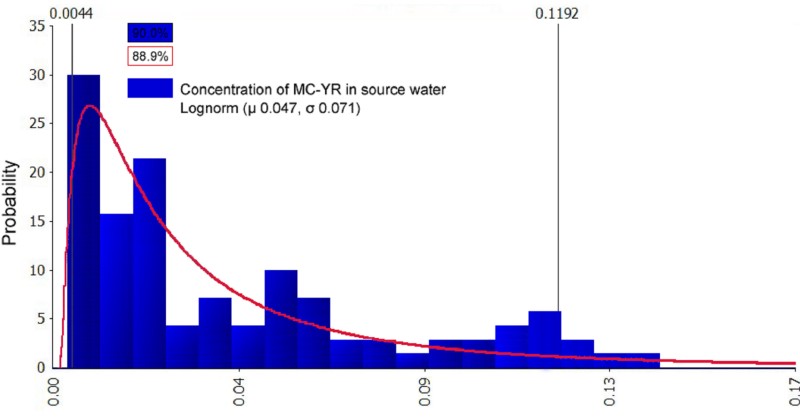

**Figure 5 Probability distribution graph after fitting of MC-YR in source water (µg/L).** The data comparison showed that the best-fitted distribution of MC-YR concentration was Lognorm (µ 0.047, σ 0.071). The abscissa in Fig. 5 represents the concentrations of MC-YR; the concentrations are partitioned, the length of each interval is the group distance, the rectangular area is the frequency of the group, and the ratio of the total sample utilized, and the vertical axis is the frequency divided by the group distance obtained. The fitting results can be visually observed from the coincidence of the blue rectangular shape (90.0%) and the area under the red curve (88.9%).

intake and the non-carcinogenic hazards index of P90 and P95 at high level of exposure was <1. This suggested that MCs, which are ingested through drinking water, pose less risk to health. The non-carcinogenic HI of MC-RR, MC-YR, and MC-LR increased successively, indicating that MC-LR was more hazardous to human health than MC-YR and MC-RR. The MC-RR, MC-YR, and MC-LR display a non-carcinogenic index in more children than adults; thus, MCs are detrimental to children.

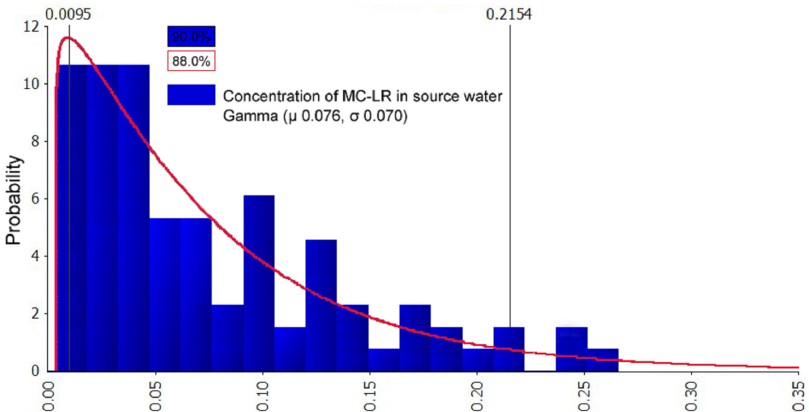

**Figure 6 Probability distribution graph after fitting of MC-LR in source water (µg/L).** The data comparison showed that the best-fitted distribution of MC-LR concentration was Gamma (µ 0.076, σ 0.070). The abscissa in Fig. 6 represents the concentrations of MC-LR; the concentrations are partitioned, the length of each interval is the group distance, the rectangular area is the frequency of the group, and the ratio of the total sample utilized, and the vertical axis is the frequency divided by the group distance obtained. The fitting results can be visually observed from the coincidence of the blue rectangular shape (90.0%) and the area under the red curve (88.0%).

**Table 6 Daily exposure to MCs intake through drinking water (µg/kg/d).**

| Projects | MC-RR | | MC-YR | | MC-LR | |
|---|---|---|---|---|---|---|
| | Adult | Child | Adult | Child | Adult | Child |
| Mean | 0.002 | 0.005 | 0.001 | 0.003 | 0.002 | 0.005 |
| P50 | 0.002 | 0.004 | 0.001 | 0.001 | 0.002 | 0.004 |
| P85 | 0.004 | 0.008 | 0.003 | 0.006 | 0.004 | 0.009 |
| P90 | 0.004 | 0.009 | 0.003 | 0.007 | 0.005 | 0.010 |
| P95 | 0.005 | 0.010 | 0.003 | 0.007 | 0.006 | 0.013 |

**Table 7 Carcinogenic exposure of MCs from source water.**

| Projects | MC-RR | | MC-YR | | MC-LR | |
|---|---|---|---|---|---|---|
| | Adult | Child | Adult | Child | Adult | Child |
| Mean | $1.27 \times 10^{-5}$ | $1.37 \times 10^{-5}$ | $3.34 \times 10^{-6}$ | $5.76 \times 10^{-6}$ | $5.20 \times 10^{-6}$ | $8.10 \times 10^{-6}$ |
| P50 | $1.40 \times 10^{-5}$ | $1.43 \times 10^{-5}$ | $2.21 \times 10^{-6}$ | $4.40 \times 10^{-6}$ | $4.69 \times 10^{-6}$ | $8.30 \times 10^{-6}$ |
| P85 | $5.43 \times 10^{-5}$ | $1.13 \times 10^{-4}$ | $6.93 \times 10^{-6}$ | $1.09 \times 10^{-5}$ | $9.18 \times 10^{-6}$ | $1.28 \times 10^{-5}$ |
| P90 | $5.47 \times 10^{-5}$ | $1.45 \times 10^{-4}$ | $7.60 \times 10^{-6}$ | $1.16 \times 10^{-5}$ | $9.85 \times 10^{-6}$ | $1.32 \times 10^{-5}$ |
| P95 | $8.63 \times 10^{-5}$ | $1.83 \times 10^{-4}$ | $8.11 \times 10^{-6}$ | $1.20 \times 10^{-5}$ | $1.11 \times 10^{-5}$ | $1.38 \times 10^{-5}$ |

## DISCUSSION

The WHO established a guide value of one µg/L for MC-LR concentration in drinking water (*World Health Organization (WHO), 2017*). A comprehensive Australian report shows that the concentration of total MCs in drinking water should not exceed 1.3 µg/L, expressed as MC-LR toxicity equivalents. Furthermore, a cell density of

**Table 8 Maximal acceptable level and negligible level recommended by different institutions.**

| Institutions | Maximum risk level | Ignore the level of risk | Remarks |
|---|---|---|---|
| USEPA | $1 \times 10^{-4}$ | – | Radiation |
| ICRP | $5 \times 10^{-5}$ | – | Radiation |
| Royal Association of England | $1 \times 10^{-6}$ | $1 \times 10^{-7}$ | – |
| Holland Environmental Protection Agency | $1 \times 10^{-6}$ | $1 \times 10^{-8}$ | Chemical contaminants |
| Swedish Environmental Protection Agency | $1 \times 10^{-6}$ | – | Chemical contaminants |

**Table 9 Non-carcinogenic exposure risk of MCs using source water.**

| Projects | MC-RR | | MC-YR | | MC-LR | |
|---|---|---|---|---|---|---|
| | Adult | Child | Adult | Child | Adult | Child |
| Mean | 0.005190 | 0.011352 | 0.030260 | 0.066190 | 0.054490 | 0.119200 |
| P50 | 0.004123 | 0.009019 | 0.017090 | 0.037380 | 0.040400 | 0.088380 |
| P85 | 0.009157 | 0.020031 | 0.067480 | 0.147610 | 0.104530 | 0.228660 |
| P90 | 0.009722 | 0.021267 | 0.077190 | 0.168840 | 0.118790 | 0.259860 |
| P95 | 0.011903 | 0.026038 | 0.085150 | 0.186270 | 0.153860 | 0.336580 |

approximately 6,500 cells/mL (biovolume of 0.6 mm$^3$/L) would be equivalent to the guideline of 1.3 µg/L MC-LR toxicity if the toxin was fully released into the water (*NHMRC & NRMMC, 2011*). In the study of a Canadian group, the recommended MAL in drinking water is 0.5 µg/L of MC-LR, or in the absence of potency equivalency values for other MCs, one µg/L of total MCs (*Watanabe et al., 1996*).

By monitoring of the water quality in the Yongjiang river, we demonstrated that although no major algal bloom occurred, MC-RR, MC-YR, and MC-LR were present in the water column during the monitoring period. The TMCs concentrations varied from 0.0313 to 0.4585 µg/L. An earlier study of MCs in Guangxi showed that the average concentration of MCs in source water and treated water supplies were 0.277 µg/L and 0.221 µg/L, respectively (*LV et al., 2005*). Another survey showed that the concentration of MCs in the source water of high-incidence areas of liver cancer in Guangxi was 15.64 ± 2.08 ng/L, and the concentration in treated water supplies was 14.42 ± 2.28 ng/L (*Li et al., 2016*). These results suggest that MCs are detected in parts of Guangxi, but without considering the influencing factors and health risk assessment of MCs. The collected data indicated that the peak level in October followed by a sharp drop in concentration when using TMCs content as an indicator (Fig. 3). The significant decrease in TMCs content may result from the decreasing of temperature from November to December, and result in a slow growth of *Microcystis*. These phenomena were similar to those described by previous groups in Tai, Yang-cheng, and Xuanwu lakes in China (*Xu et al., 2010*; *Li, Gu & He, 2014*). TMC-RR concentration reached maximum level in October and then decreased to an average concentration level in November. As compared to the concentration of the above two toxins, TMC-YR concentration was

the lowest of the three toxins studied; these results were identical to findings by other researchers, which suggested that the MCs are primarily dominated by TMC-RR and TMC-LR (*Yang et al., 2006*; *Bi et al., 2017*). It was clearly shown that the concentration of TMC-LR gradually increased from September to November. Such variation may be influenced by the differences in nutrients and climates, which are in favor of TMC-RR, TMC-LR and to a lesser extent TMC-YR.

Previous studies demonstrated that the algal toxins are produced by algae and consequently the concentration of toxins in water depends mainly on algal abundance (or biomass) such as chlorophyll-a concentration or algal cell counts, which in turn, is regulated by the environmental factors. The relationship of physical and chemical water parameters to the concentration levels of toxins are shown in Table 2. In this study, it was evident that temperature was positive and significantly correlated with concentration levels of TMCs, TMC-LR, TMC-RR, and weakly associated with concentration levels of TMC-YR. The highest concentrations of TMCs and TMC-RR, TMC-LR were observed in October and November with surface water temperature were around 25.6 °C and 26.2 °C, respectively. When water temperature increased, even higher concentration of TMCs and TMC-RR, TMC-LR concentration were detected. These findings are in agreement with previous reports which showed the concentrations of TMCs and TMC-RR, TMC-LR were temperature dependent, and TMC-RR which are generally detected at lower temperatures as compared to TMC-LR which favors at higher temperatures (*Wang et al., 2010*; *Mantzouki et al., 2018*). Intriguingly, the pH value was also shown to be related to the concentration level of toxins in the Yongjiang river. The maximum toxin concentration was detected at a pH below or above the medium level. As a result, TMC-RR and TMCs were negatively correlated with pH value (Table 2), which was similar to the results of other studies. Notably, the phytoplankton is known to affect the pH, and then, further affects the concentration levels of toxins. Therefore, the pH value cannot be used as an appropriate parameter to determine the concentration levels of toxins. A majority of the blue-green algae can grow adequately in the water at pH of 6.5–7.9 (*Wang et al., 2002*). The pH of Yongjiang river was within this range. DO concentration ranged from 2.0 to 12.5 mg/L during the study period. The reported environmental standard for river water is five mg/L (*Guobiao (GB) Chinese National Standard, 2002*). DO of Yongjiang river was partially lower than the reported standard during the monitoring period; these results show that the water is contaminated by organic matter, the oxygen consumption is severe, DO cannot be replenished in time, and the anaerobic bacteria in the water will multiply quickly (*Wang et al., 2002*). DO showed a negative correlation of TMC-LR with TMCs in Yongjiang river. However, some studies indicated that increases in oxygen saturation were correlated with algal biomass (*Bi et al., 2017*). Nonetheless, the algal abundance (or biomass) such as chlorophyll-a concentration or algal cell counts was not measured, and thus DO has no direct effect on the concentration levels of toxins. The correlation analysis results indicated that increasing the TP concentration could increase the concentration levels of toxins, especially that of TMC-LR. The current observations were in agreement with those from a study conducted in the large eutrophic Lake Erie in the USA (*Harke et al., 2016*), which demonstrated

positive correlations between TP and the abundance of toxic Microcystis and MCs. Consistent with the trend, *Vézie et al. (2002)* also found that higher *P* concentrations were beneficial to the growth of toxic *Microcystis*. Although TP was a dominant explanatory variable, the effect of TN on the concentration levels of toxins could not be ignored. The concentration of TMC-YR was negatively correlated with the TN/TP ratio. Previous studies also demonstrated that decreasing the TN/TP ratio concentration could promote the growth and toxin concentration of Microcystis (*Yu et al., 2014*; *Lei et al., 2015*). According to stepwise multiple linear regression (Table 3), TP was found to be the dominant factor affecting the contents of TMC-LR and TMCs, and pH and TN/TP ratio as the main factors affecting the content of TMC-RR and TMC-YR. These findings were in agreement with the results of correlation analysis.

The Monte Carlo simulation model determined the risk level and putative human exposure scenarios associated with the blooms in the Yongjiang river used for drinking. The whole process of security risk assessment was always accompanied by the uncertainty. The entire process of risk assessment was conducted in two steps: exposure assessment and hazard characterization. Although the extrapolation of the experimental results does not lead to certainty, it could be carried out from experimental animals to the general population and from the general population to specific populations (sensitive populations). The variations in human individuals involved parameters such as genetics, age, sex, environment (nutritional status), and other factors. On the other hand, missing data or limitations led to uncertainties, including NOVEL, time differences, and lack of exposure data. Recent studies have gradually established superior methods, such as benchmark dose (BMD) and chemical-specific adjustment factor (CSAF), to address and reduce the uncertainty in the risk assessment (*Ibelings et al., 2015*). The USEPA and Health Canada have gradually started utilizing the BMD and CSAF methods to develop the health guidance values (*Zeller, Duran-Pacheco & Guerard, 2017*).

Several countries that regulate cyanotoxins in drinking water use a parametric value based on the WHO Guidelines for one µg/L MC-LR (*World Health Organization (WHO), 2017*). With respect to the drinking source waters, most countries use guideline values based on cyanobacterial biomass (cell density, chlorophyll-a, biovolume) indirectly reflecting the potential hazardous MCs concentrations (*Valerio et al., 2009*; *Menezes, Churro & Dias, 2017*). Our results indicated that the risk of carcinogenicity of MC-RR to children health under high exposure was greater than the maximum acceptable risk level recommended by USEPA ($1 \times 10^{-4}$); the annual risk of carcinogenic exposure in adults with MC-RR was greater than the maximum acceptable risk level recommended by the ICRP ($5 \times 10^{-5}$). The TMC-RR concentrations varied from 0.0224 to 0.3783 µg/L during the monitoring period. Therefore, the guideline value of TMC-RR in the Yongjiang river should be <0.3783 µg/L, so as not to pose a health risk to humans. Furthermore, we must take into account the increasing usage of the Yongjiang river, not only for the production of drinking water, but also for ludic activities, such as water sports, fishing, sailing, and swimming. Thus, the relevant departments must attach great importance to the potential risks associated with the Yongjiang river in order to protect the health of their users.

## CONCLUSION

This study analyzed the influencing factors and the health risk assessment of MCs by Monte Carlo simulation in the Yongjiang river, China. The results showed that TP content may be related to TMC-LR and TMCs concentration, while pH and TN/TP ratio may be related to TMC-RR and TMC-YR concentration, respectively. The health risk assessment results showed that the risk of MC-RR for human health hazards is higher than that of MC-LR and MC-YR, and children are more vulnerable to MCs contamination than the adults. The risk of carcinogenicity of MC-RR to children health under high exposure was greater than the maximum acceptable risk level recommended by the USEPA. The annual risk of carcinogenic exposure in adults with MC-RR was greater than the maximum acceptable risk level recommended by the ICRP. The non-carcinogenic HI for MCs was <1. Therefore, MCs in the water bodies should be monitored with regarding to the carcinogenic risk to human health.

## ACKNOWLEDGEMENTS

The authors appreciate the Affiliated Tumor Hospital of Guangxi Medical University for their support in this study.

### Funding

This study was supported by grants from the National Natural Science Foundation of China (Project No. 81660561 and 81260319). The funders had no role in study design, data collection and analysis, decision to publish, or preparation of the manuscript.

### Grant Disclosures

The following grant information was disclosed by the authors:
National Natural Science Foundation of China: Project No. 81660561 and 81260319.

### Competing Interests

The authors declare that they have no competing interests.

### Author Contributions

- Chan-Chan Xiao conceived and designed the experiments, performed the experiments, analyzed the data, prepared figures and/or tables, authored or reviewed drafts of the paper, approved the final draft.
- Mao-Jian Chen performed the experiments, prepared figures and/or tables, approved the final draft.
- Fan-Biao Mei performed the experiments, approved the final draft.
- Xiang Fang performed the experiments, approved the final draft.
- Tian-Ren Huang contributed reagents/materials/analysis tools, approved the final draft.
- Ji-Lin Li contributed reagents/materials/analysis tools, approved the final draft.

- Wei Deng conceived and designed the experiments, analyzed the data, authored or reviewed drafts of the paper, approved the final draft.
- Yuan-Dong Li conceived and designed the experiments, analyzed the data, authored or reviewed drafts of the paper, approved the final draft.

## Data Availability

The raw data are provided in Supplemental Files 1 and 2.

## Supplemental Information

Supplemental information for this article can be found online at http://dx.doi.org/10.7717/peerj.5955#supplemental-information.

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
