# Peer review of "Influencing factors and health risk assessment of microcystins in the Yongjiang river (China) by Monte Carlo simulation"

_PeerJ, doi:10.7717/peerj.5955_

## Round 0.1 · original submission · Major Revisions

I agree with both reviewers in that there is some concern regarding identification of microcystins based simply on retention times. Please address this point specifically in your response. In addition to some of the technical issues raised by both reviewers, please give some attention to grammar in order to make your revised manuscript more clear. Also please note that both reviewers returned annotated manuscripts.

Reviewer 1 ·

Basic reporting

English could be improved, especially in the introduction, to enhance the flow of the manuscript.

Basic structure is ok, figures and tables are clear.

Experimental design

I cannot judge the quality or interpretation of the statistical analyses and the risk assessment part, as this is beyond my expertise. Having said that, I find the approach taken an interesting one.

Some comments on the parts that I could review:

1) The analytical method employed is not very specific, it only relies on retention times. As seen in Figure 1, there is a retention time difference between MC-LR standard and what is assigned to be MC-LR in a sample. As the other two compounds show no retention time shift, additional data (e.g. standard addition of a sample) should be taken to show that MC-LR is correctly identified. As the data are shown now, the last peak in figure 1B might as well be another MC. In addition, I would like to see some more information on the analytical method, e.g. recovery and limit of detection/quantification.

Validity of the findings

1) MC-RR and MC-LR occur in similar concentrations. MC-RR is generally acknowledged to be about ten times less toxic than MC-LR (confirmed by authors in line 196). Therefore, it is surprising to me that the authors find that MC-RR poses the highest health risk. This needs to be either explained or rectified. In the abstract, it should also made clear which health risks (carcinogenic or non-carcinogenic) are discussed.

2) The correlation of MCs and environmental factors is discussed, but interpreted as causation (e.g. line 342). This should not be done! Phytoplankton is known to influence pH (dense blooms increase pH) and DO (dense blooms enhance the daily DO fluctuations). So in these types of study, causation can not be proven, at best correlation can be identified.

Additional comments

Please see the attached pdf for additional small comments

Annotated reviews are not available for download in order to protect the identity of reviewers who chose to remain anonymous.

Reviewer 2 ·

Basic reporting

Use of the English language should be improved to ensure your international audience can clearly understand your text. I suggest you have a native English speaking colleague review your manuscript, in particular the discussion section. In this section it is difficult to follow the development of ideas and arguments. Some examples where the language could be improved include lines: 43-44, 104-105, 161-165, 178-187, 226, 323-326, 328-331, 332-338.
The references cited are from academically appropriate sources, mainly reviewed journals. However, it is important to avoid citing indirect references. The main problem with the references in this paper is that many of them are not accurate or relevant to support the stated argument. Some examples where references lack relevance and may be replaced by more accurate ones include lines: 45, 46, 54, 112, 118, 309, 311-312, 314.
Format of in-text citations and references should be also revised. Some examples: lines 54, 61, 211. In particular, the reference list should include the full list of authors with initials.
The structure of the article conforms to the standard format suggested by the journal. However, I suggest that section “3.1 Analysis of MCs by HPLC” (lines 209-215) be included in sub-section “2.4 Determination of MCs” within section “2. Materials and Methods” as it describes the conditions and not the results of the analyses.
Figures are relevant to the content of the article and they have sufficient resolution. In the axis labels, please consider writing the units in brackets, i.e. (µg L-1) instead of using backslash-unit “/µg L-1”. In the description of Figure 2, please consider clarifying that TMCs is the sum of individual totals (intra and extracellular) MC-LR, MC-RR and MC-YR. For figures 3, 4 and 5 please see specific comment in the manuscript (line 260).
Regarding the tables, in general they are clear and well described. When possible, It would be helpful if the author showed the monthly microcystin concentration with the associated standard deviation. Please see some specific comments in lines 221 and 260 of the manuscript.
Regarding the data, I thank you for providing the raw data; however, your supplemental files need more descriptive metadata identifiers to be useful to readers. For example, for supplemental data No. 1 it would be useful if you identified the data set by monitoring station. Also, in supplemental data No. 2, it is important that you write the dates when the environmental records were taken.

Experimental design

This research is within the Area of Environmental Sciences; as such, it is within the aims and scope of the journal. In relation to the research question, the paper clearly defines the research gap.
Regarding the research gap, this research is relevant and meaningful for the geographical study area. It contributes information on the presence, concentration, temporal variability and risk assessment of algal toxins (microcystins) in a river which is the source of water for the city of Nanning. These types of studies, where concentrations of algal toxins are associated to environmental factors, have been carried out worldwide. In this case, the importance lies in its local value, as it seems from the international literature that it is the first time that these types of toxins and their risk to human health are evaluated in this area.
In relation to the performance of the investigation, this investigation addresses two main ideas. The first idea is the identification and quantification of three types of microcystins (LR, RR and YR) in river water and the evaluation of their temporal variability in relation to environmental factors such as water temperature, pH, etc. The second idea of this paper is related to evaluating the risk of the presence of these toxins on human health by using a simulation technique (Monte Carlo Simulation). In relation to the first idea, the article (with some methodological weaknesses described below) demonstrates the presence of these toxins in water and its association with environmental parameters. The main point to be improved in this idea is the lack of clear acknowledgment by the authors that algal toxins are produced by algae and consequently the concentration of toxins in water depends mainly on algal abundance which, in turn, is regulated by environmental factors. Even though there is some association between toxin concentrations and environmental factors (reflected in the correlation coefficient), these factors are not necessarily the cause of the variability of toxins in water but mainly factors that regulate algal growth. It would improve your introduction and discussion if you acknowledge that the abundance of toxins depends on the presence and abundance of toxic strains within the cyanobacterial population and that environmental factors are important in controlling expression of the toxin synthetase genes.
A more rigorous investigation should have included some indicator of algal abundance (or biomass) such as chlorophyll-a concentration or algal cell counts. To evaluate how environmental factors affect the toxin levels in a natural environment, it would be more accurate to correlate for example the toxin quota (i.e. the concentration of toxins per cell or per unit of biomass) with the environmental factor. Also, data on the type of algae (genus) present in the water would have been useful, in particular to support some arguments in the discussion section.
If no indicator of algal biomass was measured, then this should be clearly acknowledged and considered in the discussion. Please see comments in lines: 321-331, 334-336, 342.
The second idea of this paper is related to evaluating the risk of the presence of these toxins on human health by using a simulation technique (Monte Carlo Simulation). As this is not my field of expertise, I cannot comment on the technical quality of the investigation regarding this point. However, I cannot help wondering whether this assessment is necessary considering that the concentrations of toxins (MC-LR) found in the river water are well below (range: 0.0341-0.2663 µg L-1) the guideline value suggested by the WHO in drinking water (1µg L-1) and by the USEPA (ten-Day Drinking Water Health Advisories : bottle-fed infants=0.3 µg L-1, school age children and adults=1.6 µg L-1). Maybe some more background and discussion clarifying this idea would be helpful. Lastly, there are no ethical issues to be addressed in this research.

Regarding the methods, as I mentioned earlier, the paper addresses two main ideas which are interconnected; however, each of these requires a different methodological approach. Regarding the methodology of the first idea: both sampling and water analyses need to be better referenced (please refer to comments in the manuscript). I highly recommend the authors to contact the laboratory which performed the analyses and ask for the methodology and cite the methodology. The cited references are not relevant to describe the methodology. The methodology for microcystin detection, on the other hand, has been described in greater detail. Nevertheless, I recommend the authors to have the English revised as it difficult to follow. Also, the reference cited for microcystin detection is not in the reference list. Please see specific comments in the manuscript (lines 112, 115 and 118).
Regarding the statistical analyses performed: The use of Pearson’s correlation and linear regression requires a validity check for certain assumptions (e.g. linearity, normality). I recommend the authors to mention whether they checked the assumptions or whether they performed some kind of variable transformation before applying the statistical methods. In general, environmental variables do not present normal distribution.
Regarding the methodology of the second idea (Monte Carlo Simulation): as this is not my field of expertise, I cannot comment on technical details of the method. However, the description of the methods (line 137-201) is difficult to follow mainly due to faulty English and typos.

Validity of the findings

I thank the authors for providing the raw data. A valuable aspect of the microcystin data analysis is the use of triplicates to calculate a monthly value. However, the data concentration presented in Fig. 1, Table 1 and the supplemental material No. 1 have some inconsistencies. First, data presented in supplemental data No. 1 do not match the data presented in Fig. 1 and Table 1. Microcystin concentration values in the supplemental data are 1000 times higher than those presented in Fig. 1 and Table 1.
Second, the concentrations of microcystins presented in Fig. 1 and Table 1 are below 1µg L-1. The detection limit of the technique used by the researchers has not been mentioned; however, to my knowledge, the detection limit of microcystins using HPLC-UV is generally 1µg L-1, therefore values lower than 1µg L-1 would not be detectable. I assume the authors did some calculation to account for the concentration step during the pre-treatment of the samples (i.e. they calculated the final concentration value applying a dilution factor). These two problems lead me to believe that there may be a calculation problem. I recommend the authors to verify this.
I also recommend the authors to consider the significant digits when reporting the concentrations based on the precision of the equipment and the inclusion of the SD
This is important, in particular, when comparing concentration levels. It should be clear that the differences between values are not due to the uncertainty of the measurement but due to real differences. As the authors used triplicates for the estimation of the monthly value I would recommend them to include standard deviations. Regarding the discussion and conclusions, they are linked to the original research questions; however, the English should be revised as language misuse makes it difficult to fully grasp the meaning. Please refer to earlier comments and my comment in the manuscript (line 378).

Additional comments

No comment

Annotated reviews are not available for download in order to protect the identity of reviewers who chose to remain anonymous.

---

## Round 0.2 · Minor Revisions

Thank you for your efforts to revise your manuscript. Unfortunately, the grammar is still not good. I have attached a pdf of your revised manuscript. I have edited the first 2+ pages as an example for you to work from or to return it to the service you indicated you used in your response to previous reviewer comments. I also note that it your rebuttal letter needs grammar editing, as it is difficult to understand your response to review comments.

---

## Round 0.3 · accepted · Accept

Thank you for your efforts to improve the readability of your manuscript.

#